# Novel Energetic Co-Reactant for Thermal Oxide Atomic Layer Deposition: The Impact of Plasma-Activated Water on Al_2_O_3_ Film Growth

**DOI:** 10.3390/nano13243110

**Published:** 2023-12-10

**Authors:** João Chaves, William Chiappim, Júlia Karnopp, Benedito Neto, Douglas Leite, Argemiro da Silva Sobrinho, Rodrigo Pessoa

**Affiliations:** 1Laboratório de Plasmas e Processos, Departamento de Física, Instituto Tecnológico de Aeronáutica, Praça Marechal Eduardo Gomes 50, São José dos Campos 12228-900, Brazil; julia_karnopp@outlook.com (J.K.); botan.bdn@gmail.com (B.N.); leite@ita.br (D.L.); argemiro@ita.br (A.d.S.S.); 2Laboratório de Plasmas e Aplicações, Departamento de Física, Faculdade de Engenharia de Guaratinguetá, São Paulo State University (UNESP), Guaratinguetá 12516-410, Brazil; chiappimjr@yahoo.com.br

**Keywords:** plasma-activated water, atomic layer deposition, alumina, growth per cycle

## Abstract

In the presented study, a novel approach for thermal atomic layer deposition (ALD) of Al_2_O_3_ thin films using plasma-activated water (PAW) as a co-reactant, replacing traditionally employed deionized (DI) water, is introduced. Utilizing ex situ PAW achieves up to a 16.4% increase in the growth per cycle (GPC) of Al_2_O_3_ films, consistent with results from plasma-enhanced atomic layer deposition (PEALD). Time-resolved mass spectrometry (TRMS) revealed disparities in CH_4_ partial pressures between TMA reactions with DI water and PAW, with PAW demonstrating enhanced reactivity. Reactive oxygen species (ROS), namely H_2_O_2_ and O_3_, are posited to activate Si(100) substrate sites, thereby improving GPC and film quality. Specifically, Al_2_O_3_ films grown with PAW pH = 3.1 displayed optimal stoichiometry, reduced carbon content, and an expanded bandgap. This study thus establishes “PAW-ALD” as a descriptor for this ALD variation and highlights the significance of comprehensive assessments of PAW in ALD processes.

## 1. Introduction

Plasma-activated water (PAW) has gained widespread attention in recent years due to its increased chemical reactivity, which is achieved through the transfer of energy from a gaseous plasma to water without the use of any additional chemicals. Its versatility extends to fields like agriculture, medicine, and dentistry, thanks to its notable biochemical properties. In agriculture, PAW boosts seed germination and plant growth [1]. In medicine and dentistry, it is harnessed for cancer therapy, wound healing, biofilm removal, disinfection, and teeth whitening [2,3,4]. Furthermore, PAW augments food safety and quality, showcasing its adaptability in various applications [5,6]. The pronounced biochemical efficacy of PAW stems from the reactive oxygen and nitrogen species (RONS) present, such as nitrous acid (HNO_2_), hydrogen peroxide (H_2_O_2_), nitrite (NO_2_^−^), nitrate (NO_3_^−^), and ozone (O_3_) [2]. While their concentrations are low, their prolonged existence magnifies their impact. Post plasma activation, the RONS concentration in PAW can spike to several hundred milligrams per liter, accounting for less than 1% of the total volume [2,6,7], underscoring the potent effects of these species.

The utilization of PAW in life sciences is well established, yet its application in nanotechnology remains nascent, with only a handful of studies exploring its potential. P. Galář et al. demonstrated the treatment of silicon nanoparticles with PAW, leading to a significant increase in the photoluminescence quantum yield of these particles, as well as a notable improvement in their water dispersibility [8]. In another study, N. Sharmin et al. synthesized silver nanoparticles averaging 22 nm in size [9]. They employed the reactive nitrogen and oxygen species in PAW as reducing agents within a sodium alginate (SA) solution, which simultaneously acted as a stabilizer for the nanoparticles. However, in the specific area of nanostructured thin film growth through chemical or physical deposition methods, no research has been reported to date. 

To expand the scope of PAW applications, this study explores the use of PAW as a co-reactant in the thermal ALD of Al_2_O_3_ thin films using trimethylaluminum (TMA) reactant. We activated DI water using a non-thermal plasma jet, then we introduced PAW to the ALD reactor’s co-reactant line. Al_2_O_3_ thin films were grown on Si(100) using PAW and activated for 4, 30, and 60 min, resulting in pH values of 3.5, 3.1, and 2.7, respectively. The concentration of RONS in each PAW sample was measured using UV-Vis absorption spectroscopy. To understand the surface chemisorption dynamics of RONS in PAW interacting with TMA, an in situ analysis was performed with TRMS. Additionally, thin films of Al_2_O_3_, derived from both deionized (DI) water and PAW, were subjected to comprehensive characterization. This involved mechanical profilometry, Fourier-Transform Infrared Spectroscopy (FT-IR), and X-ray Photoelectron Spectroscopy (XPS) to assess their properties.

## 2. Materials and Methods

### 2.1. PAW Synthesis and Characterization

PAW was prepared using a gliding arc plasma jet (GAPJ) at a flow rate of 5 L/min and a discharge power of 7 W. Additional details about GAPJ are available in the Appendix A. DI water (40 mL), boasting a resistivity of 10 µS/cm and a pH of 6.7, was positioned 0.3 cm from the GAPJ reactor nozzle, followed by plasma activation for 4, 30, and 60 min. The ex situ PAW’s physicochemical parameters were assessed with a multi-parameter water meter (Metrohm 913 phmeter, São Paulo, Brazil), and the RONS (H_2_O_2_, NO_2_^-^, NO_3_^-^, and HNO_2_) concentrations were ascertained through UV-Vis spectrophotometry (Thermo Fisher Scientific Inc., Waltham, MA, USA). The absolute concentrations of the RONS were determined using the procedure described in [10]. To determine the O_3_ concentration, the multiparameter photometer Exact Micro 20 (Industrial Test Systems, Rock Hill, SC, USA) was used. See Table 1 for the results.

### 2.2. Al_2_O_3_ Thin Film Deposition and In Situ Process Monitoring

After generation, each PAW was transferred into the cross-flow-type thermal ALD reactor as a co-reactant. The ALD cycle had four steps: 0.15 s with TMA (97%), 4.00 s N_2_ purge, 0.30 s of PAW or DI water, followed by another 4.00 s N_2_ purge. The reactor’s base pressure was below 10^−2^ mbar, with the deposition pressure maintaining around 0.5 mbar using 100 sccm of N_2_. Al_2_O_3_ films were cultivated on p-type Si(100) wafers at 150 °C, with 100 to 1500 reaction cycles for each PAW and DI water.

In this study, the TRMS was employed to track the primary species produced in each ALD pulse: methane (CH_4_^+^, 16 amu), water (H_2_O^+^, 18 amu), and ethane (C_2_H_6_^+^, 30 amu). The partial pressure of the surface reaction by-product species within each pulse was analyzed using a mass spectrometer (RGA-200, Stanford Research Systems, Sunnyvale, CA, USA), configured to the reactor exhaust to detect residual species via a micro-orifice.

### 2.3. Material Characterization

The thickness of the Al_2_O_3_ films was gauged using a KLA Tencor P-7 profilometer, by creating a step in specific regions of the samples using Kapton tape. The samples’ chemical bonding was examined with infrared measurements via an ATR-FTIR PerkinElmer 400 IR spectrometer at a 2 cm^−1^ resolution. Additionally, the films underwent characterization by XPS with a K-Alpha Thermo Scientific spectrometer (X-ray Al-Kα, hν = 1486.6 eV) under pressures less than 10^−7^ mbar. The pass energy applied to obtain the spectra was 200 eV for survey and 50 eV for high-resolution scans.

## 3. Results and Discussion

Table 1 shows a comprehensive overview of the changes in key parameters of PAW during the activation process. Notably, there is a significant decrease in pH over time, indicating increasing acidity, a common characteristic of PAW. This drop in pH is paralleled by marked changes in other properties. The ORP increases, reflecting a more oxidizing environment within the water. Conductivity and TDS both show a significant uptick. The increase in conductivity suggests a greater ion concentration, which is corroborated by the rise in TDS, indicating more dissolved substances. These changes are indicative of the complex chemical transformations occurring in PAW. 

Furthermore, the concentration of various RONS, namely H_2_O_2_, HNO_2_, NO_3_^−^, NO_2_^−^, and O_3_, also increased. H_2_O_2_ and O_3_, known for their strong oxidative properties, show a particularly notable rise. The increase in these RONS concentrations aligns with the enhanced oxidative potential of PAW, as indicated by the ORP readings. Each of these RONS plays a unique role in determining the chemical and physical properties of PAW, contributing to its potential applications, particularly in the growth of oxides through ALD.

As the XPS results did not reveal the incorporation of nitrogen into Al_2_O_3_ films (see Appendix A), the following discussions will primarily focus on reactive oxygen species (ROS), such as H_2_O_2_ and O_3_, in the ALD process using PAW as a co-reactant.

Figure 1a depicts a linear growth in the observed ALD processes with each reaction cycle, emphasizing the self-limiting nature of the ALD process when using PAW. Figure 1b demonstrates the GPC as a function of co-reactant pH. The data underscore PAW’s potential to boost the GPC of Al_2_O_3_ films. This trend likely results from the RONS produced during the plasma activation of water, leading to a GPC increase of up to 16.4% in this study.

To elucidate the influence of RONS in PAW on the Al_2_O_3_ ALD chemisorption process, in situ TRMS was utilized. This method offered valuable insights into the gaseous by-products formed during the initial 20 ALD cycles on Si(100) substrates and on developing Al_2_O_3_ layers. The observed species mainly consisted of reaction by-products, specifically from interactions between TMA and PAW at a pH of 2.7 (illustrated in Figure 2a,b), as well as TMA in conjunction with DI water (shown in Figure 2c,d). Notably, the initial introduction of TMA resulted in the detection of methane (CH_4_^+^, m/e = 16) and ethane (C_2_H_6_^+^, m/e = 30) signals. While the methane signal was present during both TMA and H_2_O exposure phases, the ethane signal was uniquely observed during TMA exposure periods [11].

It is important to note that during the initial ALD cycles on Si(100), characterized by a lack of substantial hydroxyl groups, relatively little CH_4_ is produced. The comparison between Figure 2a,c reveals that for PAW, from the second cycle onwards, the partial pressure of CH_4_^+^ exceeds that of C_2_H_6_^+^, while in the case of H_2_O, this phenomenon is only observed from the fourth cycle. As the ALD process progresses and multiple cycles are completed, Al_2_O_3_ forms a continuous film on the substrate. At this stage, TMA primarily interacts with the hydroxyl groups on the Al_2_O_3_ surface, resulting in an intensification of the CH_4_^+^ signal. Additionally, during the co-reactant pulse, the oxidant species oxidize the Al–CH_3_ surface, leading to the release of CH_4_. A comparison between Figure 2b,d reveals an increase in the partial pressure of CH_4_^+^ during the co-reactant pulse in the PAW-ALD process. 

The oxidation process in PAW-ALD appears more intricate than in thermal ALD using H_2_O. The ROS in PAW, namely H_2_O_2_ and O_3_, could introduce alternative oxidation pathways for –CH_3_ surface groups. Seo et al. investigated the molecular reactivity of Al_2_O_3_ ALD at lower deposition temperatures using three different oxidants, proposing a reactivity sequence of H_2_O < H_2_O_2_ < O_3_ [12]. Indeed, Elliott et al. found that the thickness of Al_2_O_3_ films deposited using O_3_ is temperature-dependent [13]. Their observations indicated that at 150 °C, Al_2_O_3_ film thickness using O_3_ increased by up to 18.75% compared to films grown with H_2_O as a co-reactant. Nam et al. also explored the use of O_3_ as a co-reactant at low processing temperatures (50–125 °C). However, they observed that at 125 °C, the resulting Al_2_O_3_ film was thinner compared to the one produced using H_2_O as the co-reactant [14]. In the case of H_2_O_2_, literature evidence suggests that it can easily adsorb onto surfaces, thereby enhancing surface oxidation [12,14]. Al_2_O_3_ films deposited solely with H_2_O_2_ [14,15], or a combination of H_2_O_2_ and H_2_O [16], exhibited increased thickness compared to those grown with H_2_O at 150 °C. 

These results are consistent with the pronounced CH_4_^+^ signal observed in Figure 2b. Moreover, an increase in the partial pressure of higher hydrocarbons, such as C_2_H_6_^+^, during the TMA exposure phase (as shown in Figure 2a) suggests the possibility of alternative reaction pathways, similar to those observed in PEALD [17]. The interaction between O_3_ and the –CH_3_ surface groups predominantly results in the production of CO_2_ and H_2_O (as indicated by the observed increase in H_2_O^+^ in Figure 2a when using PAW). However, this interaction can also generate CH_4_ [18,19], contributing to an elevated partial pressure of CH_4_^+^ during the PAW pulse. Similarly, the reaction of H_2_O_2_ with the surface yields CH_4_ [12], further augmenting the partial pressure of this by-product.

Figure 3 presents the FT-IR spectra of the alumina films deposited using different PAW samples. Each spectrum is characterized by a broad band ranging from 400 to 1000 cm^−1^, indicative of Al–O interactions, which are typical in both AlO_4_ and AlO_6_ structures [20]. Within this range, the peak observed around 516 cm^−1^ can be attributed to the Al–O vibrational mode in AlO_6_ [20]. Furthermore, the absorption peak near 668 cm^−1^ is largely influenced by O–Al–O bending vibrations [21].

Significantly, the pH 2.7 PAW sample displays enhanced broad features in the 600–900 cm^−1^ region, commonly associated with Al–OH bending vibrations. The increased concentration of ROS in this sample may influence the chemisorption and subsequent growth of the alumina films. This influence could result in a greater incorporation of OH groups or induce a structural change in the alumina film. Such changes are reflected in the more pronounced Al–OH stretching vibration signals observed in the spectrum.

Additionally, the absence of a peak at 530 cm^−1^, as noted by Katamreddy et al., underscores the film’s amorphous nature [22]. The consistency observed in the 1000–1400 cm^−1^ region across all samples, typically associated with the bending vibrations of adsorbed water molecules or hydroxyl groups, suggests that the surface-bound water or hydroxyl groups are not significantly affected by the varying pH levels of the PAW used. Wang et al. successfully fabricated Al_2_O_3_ films possessing the aforementioned characteristics by employing O_2_ plasma and O_3_ as co-reactants [21].

Table 2 provides an analysis of the O/Al ratio, carbon content, and bandgap (E_g_) of Al_2_O_3_ films, as determined by XPS. The film made with PAW at pH 3.1 shows an O/Al ratio close to the stoichiometric ratio and has lower carbon content than others. A consistent observation across PAW-treated films is their higher E_g_ values compared to those grown with DI water. The PAW 3.5 sample has the highest E_g_ (6.23 eV), followed by PAW 3.1 (6.09 eV) and PAW 2.7 (6.02 eV). In their recent study, Castillo-Saenz et al. determined the E_g_ of an Al_2_O_3_ thin film fabricated using O_2_ plasma, finding a value of 6.58 eV [23]. Elevated E_g_ values may suggest a uniform and dense film structure, advantageous for applications demanding enhanced electronic properties.

## 4. Conclusions

The use of PAW as a co-reactant for thermal ALD (150 °C) of Al_2_O_3_ thin films results in a GPC enhancement of up to 16.4% (with PAW 3.1) compared to the conditions with DI water. This advancement holds industrial relevance, mirroring the gains observed in the PEALD process utilizing oxygen plasma. TRMS data indicates disparities in CH_4_^+^ partial pressures during TMA reactions with DI water versus PAW. Elevated CH_4_^+^ pressures are observed in PAW reactions, pointing to enhanced reactivity. Major influencing factors include H_2_O_2_ and O_3_, which potentially activate Si(100) and subsequent Al_2_O_3_ surface sites, thereby enhancing the GPC of Al_2_O_3_. Initial material analysis suggests that Al_2_O_3_ films grown using PAW demonstrate superior quality compared to those fabricated with DI water. Notably, films derived from PAW 3.1 presented a near stoichiometric ratio and reduced carbon concentration. For a comprehensive understanding of PAW’s role in the ALD process, in-depth investigations involving different temperatures to determine the optimal ALD window, exploration of varied metal precursors, and evaluation of diverse PAWs generated by alternate non-thermal plasmas are necessary. The term “PAW-ALD” is proposed to characterize this enhanced ALD variant.

## Figures and Tables

**Figure 1 nanomaterials-13-03110-f001:**
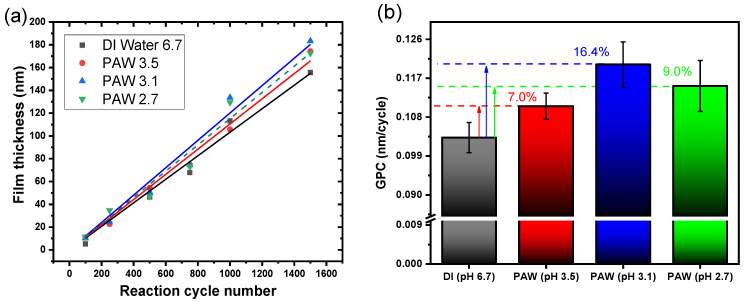
(**a**) Al_2_O_3_ film thickness as a function of reaction cycle number for various co-reactants, and (**b**) GPC in relation to the pH of the co-reactant.

**Figure 2 nanomaterials-13-03110-f002:**
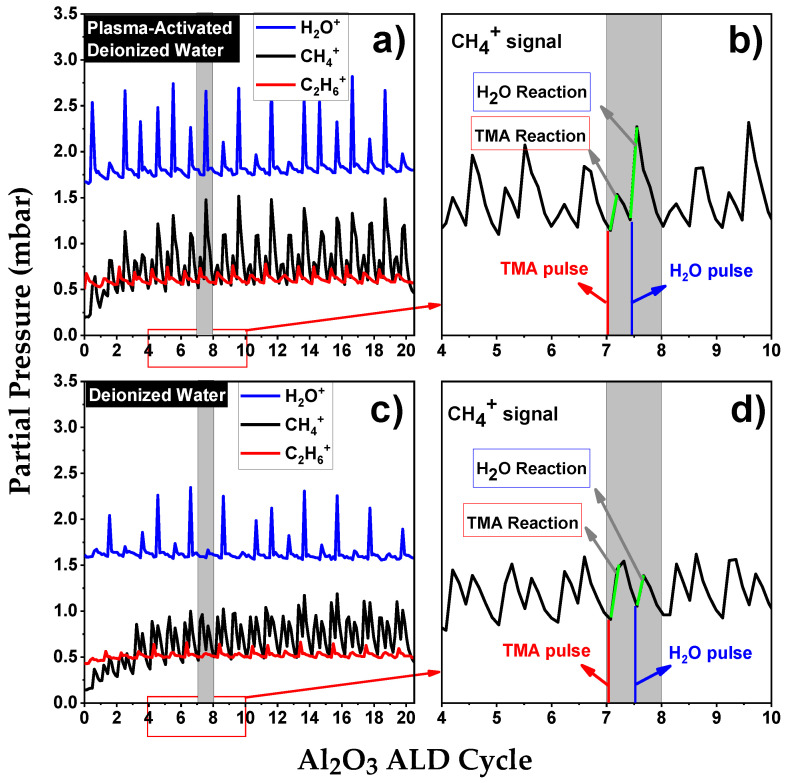
TRMS of CH_4_^+^, H_2_O^+^, and C_2_H_6_^+^ during the first 20 cycles of Al_2_O_3_ thin film growth using (**a**,**b**) PAW with pH 2.7 and (**c**,**d**) DI water. The green segment on the TRMS graph signifies the changes in partial pressure during the pulsing intervals of TMA and PAW (or DI water), respectively.

**Figure 3 nanomaterials-13-03110-f003:**
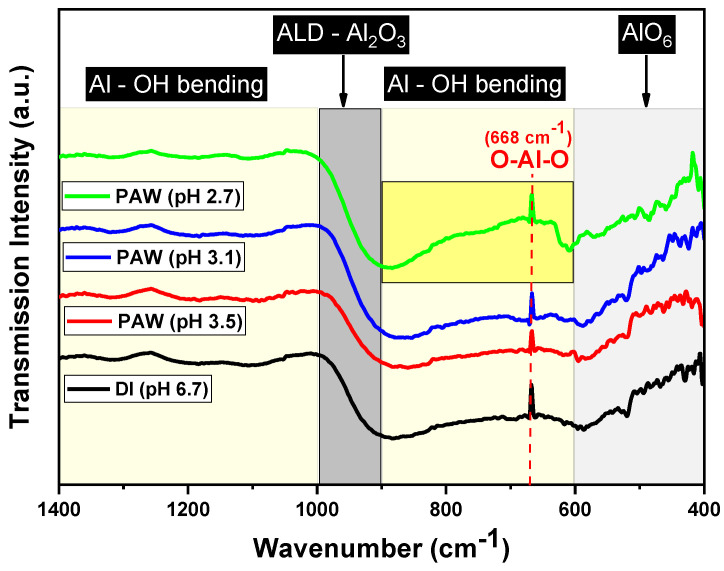
FT-IR spectra of the Al_2_O_3_ thin films deposited with DI water and PAW at different pH values.

**Table 1 nanomaterials-13-03110-t001:** Physicochemical parameters (pH, oxidation reduction potential (ORP), conductivity (σ), and total dissolved solids (TDS)) and RONS concentrations of DI water and PAWs.

Activation Time (min)	pH	ORP (mV)	σ (µS/cm)	TDS (ppm)	RONS				
					H_2_O_2_ (mg/L)	NO_3_^−^ (mg/L)	NO_2_^−^ (mg/L)	HNO_2_ (mg/L)	O_3_ (mg/L)
0	6.7	106	10	10	-	-	-	-	-
4	3.5	190	130	90	68.2	46.3	32.8	20.3	0.03
30	3.1	228	310	450	76.5	44.9	33.5	65.4	1.19
60	2.7	239	800	560	177.4	193.7	38.4	188.1	>2.00 *

* Measurement above the equipment’s detection limit.

**Table 2 nanomaterials-13-03110-t002:** O/Al ratio, C concentration, and band gap (E_g_) of the Al_2_O_3_ films.

Sample	O_o_/Al	O_t_/Al	C%	E_g_ (±0.01 eV)
DI water	1.66	1.82	3.60	5.97
PAW 3.5	1.67	1.87	3.91	6.23
PAW 3.1	1.56	1.81	3.42	6.09
PAW 2.7	1.61	1.86	3.96	6.02

## Data Availability

Data are contained within the article.

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
