# Peer review of "Novel Energetic Co-Reactant for Thermal Oxide Atomic Layer Deposition: The Impact of Plasma-Activated Water on Al2O3 Film Growth"

_nanomaterials, 2023, doi:10.3390/nano13243110_

Round 1

Reviewer 1 Report

Comments and Suggestions for Authors

The manuscript is clearly written and the figures and captions are sufficiently informative. However the results and discussion section could have more discussion and comparison to existing works to put this study in a proper context. There should also be more cross-comparison between different results and how the data can be interpreted. Therefore I recommend that the authors should revise their manuscript with necessary improvements before it can be published. My detailed comments are:

1. I am not sure how reliable the analysis of the RONS from UV-absorption spectra is.The authors have used gaussian fitting to determine the different species, however there is no reason why the absorbance could even be fitted like that, in fact even in the references they use as a basis of their analysis no gaussian peak fitting was used , but instead reference spectra of the different RON substances.Moreover, the reliability of this analysis is further hindered by the inconsistency between how the peaks are fitted (peak positions and widths)

It is fair to say based on the results that the absorbance of the pH 2.7 sample is different from the others, which is caused by different concentration of RONS, but the concentrations presented in the table 1 are very likely untrue. I would reconsider adding any quantitative data about the RONS, but to replace that with a more qualitative analysis and discussion.

2. How do the growth rates compare to the growth of ALD Al2O3 using H2O2 and O3 as an oxygen source? There are overall very little comparison to any relevant ALD literature.

3. The authors claim in their conclusions that the increase in the the GPC is due to the activation of the Si surface that boosts the growth. But this would only apply during the very first cycles as the alumina would fully cover the Si already after max. 5 cycles. In the first datapoints in Fig.1(a) all the PAW Al2O3 are essentially of the same thickness, so and the differences between the in the growth appear only later. So what is the proposed mechanism that causes the different GPC in the different PAW alumina samples?

4. In Fig3 the authors show the FTIR spectra of the grown alumina films. The spectra of  DI, pH 3.5, and pH 3.1 samples look essentially the same, but the pH 2.7 sample has enhanced broad features at 600-900 cm-1 region. How does this observation correspond to the other obtained results and why is this OH-related feature pronounced here while the other OH- associated region at 1000-1400 cm-1 is the same in all the samples? 

5. If the authors are correct and there is a significant concentration of nitrogen containing reactive species (NO2- NO3- HNO2) in the PAWs, would this mean that the films would also contain elevated concentrations of N? Was that investigated with the XPS and was any nitrogen detected in the films?

Moreover, I am not sure how definite conclusions can be drawn from the XPS results, based on the table 2 the composition in the films is almost identical (O/Al) ratios and the C%. XPS is quantitative but I would argue that the uncertainties are larger than the 2-decimal point precision that the authors present. So I would be careful from drawing any major conclusions based on this data only as the differences are very small and can be within the measurement uncertainty.

Author Response

The manuscript is clearly written and the figures and captions are sufficiently informative. However, the results and discussion section could have more discussion and comparison to existing works to put this study in a proper context. There should also be more cross-comparison between different results and how the data can be interpreted. Therefore I recommend that the authors should revise their manuscript with necessary improvements before it can be published. My detailed comments are:

  1. I am not sure how reliable the analysis of the RONS from UV-absorption spectra is. The authors have used gaussian fitting to determine the different species, however there is no reason why the absorbance could even be fitted like that, in fact even in the references they use as a basis of their analysis no gaussian peak fitting was used , but instead reference spectra of the different RON substances. Moreover, the reliability of this analysis is further hindered by the inconsistency between how the peaks are fitted (peak positions and widths)

It is fair to say based on the results that the absorbance of the pH 2.7 sample is different from the others, which is caused by different concentration of RONS, but the concentrations presented in the table 1 are very likely untrue. I would reconsider adding any quantitative data about the RONS, but to replace that with a more qualitative analysis and discussion.

Response: Thank you for your insightful comments regarding our method of analyzing Reactive Oxygen and Nitrogen Species (RONS) in plasma-activated water (PAW) using UV-absorption spectra. We appreciate your concerns about the application of Gaussian fitting for discerning different species.

In our study, Gaussian fitting was employed as a strategic approach to deconvolute the absorption spectra, facilitating the identification of RONS such as hydrogen peroxide (H2O2), nitrite (NO2-), nitrate (NO3-), nitrous acid (HNO2), and ozone (O3) using literature data. This approach was particularly utilized to highlight the presence of these RONS and to provide a detailed analysis of the UV-Vis spectra, as presented in section 2 of the supplementary material.

For measuring the absolute concentrations (in mg/L) of H2O2, NO2-, and NO3-, the methodology outlined in section 3 of our supplementary material was followed. This methodology, initially proposed by Liu et al. in 2019, has been widely adopted in various PAW studies. Some of our recent published papers used partially or totally this methodology for measure absolute concentrations of these RONS:

  1. Sampaio, A.d.G.; Chiappim, W.; Milhan, N.V.M.; Botan Neto, B.; Pessoa, R.; Koga-Ito, C.Y. Effect of the pH on the Antibacterial Potential and Cytotoxicity of Different Plasma-Activated Liquids. Int. J. Mol. Sci. 2022, 23, 13893. https://doi.org/10.3390/ijms232213893
  2. Miranda, F.S.; Tavares, V.K.F.; Gomes, M.P.; Neto, N.F.A.; Chiappim, W.; Petraconi, G.; Pessoa, R.S.; Koga-Ito, C.Y. Physicochemical Characteristics and Antimicrobial Efficacy of Plasma-Activated Water Produced by an Air-Operated Coaxial Dielectric Barrier Discharge Plasma. Water 2023, 15, 4045. https://doi.org/10.3390/w15234045

  1. How do the growth rates compare to the growth of ALD Al2O3 using H2O2 and O3 as an oxygen source? There are overall very little comparison to any relevant ALD literature.

Response: We recognize the significance of offering a thorough comparison with pertinent studies in the field of Atomic Layer Deposition, especially those employing H2O2 and O3 as oxidants. In the updated version of our manuscript, we have incorporated an expanded comparison with select studies from the literature. This comparison elaborates on both the similarities and differences in growth rates, providing a more comprehensive contextual understanding of our findings.

  1. The authors claim in their conclusions that the increase in the GPC is due to the activation of the Si surface that boosts the growth. But this would only apply during the very first cycles as the alumina would fully cover the Si already after max. 5 cycles. In the first datapoints in Fig.1(a) all the PAW Al2O3 are essentially of the same thickness, so and the differences between the in the growth appear only later. So what is the proposed mechanism that causes the different GPC in the different PAW alumina samples?

Response: Thank you for your insightful observation regarding the initial growth stages of Al2O3 films using PAW and the subsequent differences in GPC. We agree with your point that the activation of the Si surface would primarily influence the growth during the very first cycles, and that the alumina would likely cover the Si surface after approximately five cycles.

In our manuscript, while we initially attributed the increase in GPC to the activation of the Si surface, further analysis of our data, particularly from Figure 1(a), indicates that the differences in growth rates between different PAW Al2O3 samples become more pronounced in the later stages of the ALD process. This suggests that factors other than the initial activation of the Si surface are contributing to the varied GPC.

We propose that the differences in GPC observed in various PAW samples are likely due to the unique chemical properties of the PAW used, particularly the concentration and types of Reactive Oxygen and Nitrogen Species (RONS) present. These RONS, especially hydrogen peroxide (H2O2) and ozone (O3), can introduce alternative oxidation pathways for –CH3 surface groups, influencing the chemisorption process of Al2O3 ALD. The presence of these species in PAW could lead to variations in film growth characteristics, as they might interact differently with the alumina surface compared to conventional DI water.

Additionally, our observations of increased higher hydrocarbons like C2H6+ during the PAW exposure step point to potential alternative reaction pathways akin to those seen in plasma-enhanced ALD (PEALD). These alternative pathways could contribute to the differences in GPC observed in our study.

In light of your feedback, we revised our manuscript to include a more detailed discussion on the influence of RONS in PAW on the Al2O3 ALD process, especially in the later growth stages.

  1. In Fig3 the authors show the FTIR spectra of the grown alumina films. The spectra of  DI, pH 3.5, and pH 3.1 samples look essentially the same, but the pH 2.7 sample has enhanced broad features at 600-900 cm-1 region. How does this observation correspond to the other obtained results and why is this OH-related feature pronounced here while the other OH- associated region at 1000-1400 cm-1 is the same in all the samples? 

Response: Thank you for your observation regarding the FTIR spectra of the alumina films grown using different PAW samples. In Figure 3, it is indeed noted that the FTIR spectrum of the pH 2.7 PAW sample shows enhanced broad features in the 600-900 cm-1 region, which are typically associated with Al-OH stretching vibrations.

The pronounced features in this region for the pH 2.7 sample could be attributed to the higher concentration of Reactive Oxygen Species (ROS), particularly H2O2 and O3, as indicated by the quantitative data for RONS concentrations found in PAW (Table 1). These species, present in greater amounts in the pH 2.7 PAW sample, may lead to a different chemical environment during the ALD process, affecting the chemisorption and subsequent growth of the alumina films. This altered environment could result in an increased incorporation of OH groups or a change in the alumina film structure, which is reflected in the enhanced Al-OH stretching vibration signals in the FTIR spectrum.

Regarding the other OH-associated region at 1000-1400 cm-1, which appears similar across all samples, this region is typically associated with the bending vibrations of adsorbed water molecules or hydroxyl groups on the surface. The similarity in this region across all samples suggests that the surface-bound water or hydroxyl groups are not significantly affected by the varying pH levels of the PAW used. This could be due to the fact that the surface adsorption phenomena are more influenced by the physical properties of the surface rather than the chemical composition of the ALD precursors.

  1. If the authors are correct and there is a significant concentration of nitrogen containing reactive species (NO2- , NO3- , HNO2) in the PAWs, would this mean that the films would also contain elevated concentrations of N? Was that investigated with the XPS and was any nitrogen detected in the films?

Moreover, I am not sure how definite conclusions can be drawn from the XPS results, based on the table 2 the composition in the films is almost identical (O/Al) ratios and the C%. XPS is quantitative but I would argue that the uncertainties are larger than the 2-decimal point precision that the authors present. So I would be careful from drawing any major conclusions based on this data only as the differences are very small and can be within the measurement uncertainty.

Response: Thank you for your question about the potential incorporation of nitrogen into the Al2O3 films due to the presence of nitrogen-containing reactive species in the plasma-activated water (PAW).

The X-ray Photoelectron Spectroscopy (XPS) results from the study did not reveal the incorporation of nitrogen in the Al2O3 films. This was specifically mentioned in the article, stating that the XPS results did not show nitrogen incorporation in the films (refer to Figure S4 in the supplementary material)​​. Therefore, even though there was a significant concentration of nitrogen-containing reactive species (NO2-, NO3-, HNO2) in the Plasma-Activated Water (PAW), this did not result in elevated concentrations of nitrogen in the films.

Regarding the XPS measurements and values: the comparisons in this study are between samples analyzed using the same XPS equipment, which likely reduces concerns about relative uncertainties. Analyzing multiple samples under identical conditions with the same instrument lends greater reliability to the comparative results, as they are uniformly subject to the same systematic errors and calibration standards. Furthermore, the supplementary material demonstrates that the fitting procedure was executed effectively, further minimizing potential errors.

Reviewer 2 Report

Comments and Suggestions for Authors

In this manuscript, the authors investigate the effect of plasma activated water on Al2O3 growth via ALD. The authors systematically investigate the growth process based on thickness measurements, in-situ mass spectrometry, FTIR, and XPS. Their claims are supported by appropriate evidence and manuscript is well-written/organized. The manuscript also falls within the scope of nanomaterials, so I would recommend publication. I only have a couple of minor comments:

1.       I believe nitric acid is HNO3. The authors are using HNO2 throughout the manuscript. Please check.

2.       Some of the acronyms, such as ORP and TDS have not been spelled out.

3.       Figure 1 (a) suggests there is no nucleation delay, although we would always expect some sort of nucleation delay during the initial stages of growth. Have the authors tried growing with less number of cycles?

4.       I think the authors should at least outline proposed/possible reaction pathways and provide more relevant references for the sake of helping the readers.

5.       How did the authors calculate the bandgap from XPS? In general, UV-Vis spectroscopy (optical bandgap) is required in addition to the XPS/UPS measurements to understand the band structure. But, I have not seen the relevant or related information in the manuscript.

Author Response

In this manuscript, the authors investigate the effect of plasma activated water on Al2O3 growth via ALD. The authors systematically investigate the growth process based on thickness measurements, in-situ mass spectrometry, FTIR, and XPS. Their claims are supported by appropriate evidence and manuscript is well-written/organized. The manuscript also falls within the scope of nanomaterials, so I would recommend publication. I only have a couple of minor comments:

  1. I believe nitric acid is HNO3. The authors are using HNO2 throughout the manuscript. Please check.

Response: We would like to address the potential consideration of HNO₃ (nitric acid) in the context of plasma-activated water (PAW). While nitric acid is a well-known and widely studied compound, its stability in PAW is a subject of considerable complexity. PAW, characterized by a high concentration of reactive oxygen and nitrogen species (RONS), presents a highly reactive environment. This can lead to the rapid decomposition or transformation of HNO₃ into other nitrogen oxides or compounds, due to interactions with reactive species such as hydroxyl radicals, peroxides, and others found in PAW. Furthermore, the pH of PAW can significantly influence the stability of nitric acid. PAW typically exhibits a range of pH values depending on the plasma generation method and conditions. HNO₃, being a strong acid, might undergo ionization or react with other constituents in PAW under varying pH conditions, leading to reduced stability or even the formation of new compounds. Considering these factors, our study intentionally focuses on HNO₂ (nitrous acid) rather than HNO₃. HNO₂ is more representative of the nitrogen species typically found in PAW and is more relevant to the applications and phenomena we are exploring. The instability of HNO₃ in such a reactive medium as PAW underscores our choice to focus on HNO₂, which aligns with the observed chemical behavior and the practical applications of PAW in various fields, as detailed in our manuscript.

Additional Commentary: Your observation on the nomenclature is accurate; however, the consideration in these systems revolves around HNO₂. The pH is a reflection of the proton concentration within the system. When accounting for species like HNO₃, HNO₂, and H₂O₂, we focus on HNO₃ and HNO₂ as the primary contributors to the pH of plasma-activated water (PAW). This choice is informed by the relative acid dissociation constants (pKa), where pKa HNO₃ < pKa HNO₂ << pKa H₂O₂. Consequently, the proton concentration in the solution is minimally influenced by the dissociation of hydrogen peroxide.

In the case of HNO₃, being a strong acid, we assume complete dissociation, resulting in its transformation into NO₃⁻ and H+. Conversely, HNO₂ undergoes partial dissociation, meaning that, in addition to dissociating into H+ and NO₂⁻, it can also exist as HNO₂.

This rationale underscores our emphasis on HNO₂ in our study of PAW, as it is reflective of the observed chemical behavior and aligns with the practical applications in various fields, as detailed in our manuscript.

  1. Some of the acronyms, such as ORP and TDS have not been spelled out.

Response: Thank you for highlighting the oversight regarding the acronyms ORP and TDS in our manuscript. We acknowledge that these terms should have been defined at their first use for clarity. ORP stands for 'Oxidation-Reduction Potential,' which is a measure of the tendency of a chemical substance to oxidize or reduce another chemical substance. TDS stands for 'Total Dissolved Solids,' which refers to the combined content of all inorganic and organic substances contained in a liquid in molecular, ionized, or micro-granular suspended form. We have now amended our manuscript to include these definitions where these acronyms first appear, ensuring clarity for all readers. The relevant section in our manuscript can be found in Table 1, where these parameters are listed among the physicochemical parameters of PAW.

  1. Figure 1 (a) suggests there is no nucleation delay, although we would always expect some sort of nucleation delay during the initial stages of growth. Have the authors tried growing with less number of cycles?

Response: Regarding your suggestion of growing with fewer cycles to observe the nucleation phase, we did not conduct experiments with a reduced number of cycles in this particular study. Our focus was primarily on understanding the influence of RONS in PAW on the chemisorption process of Al2O3 ALD, as indicated in Figure 1b, which shows the Growth Per Cycle (GPC) in relation to the pH of the co-reactant. However, we acknowledge the importance of your suggestion and recognize that conducting experiments with a fewer number of cycles could provide additional insights into the nucleation phase, especially in the context of PAW's unique properties. We believe that such an investigation could be a valuable extension of our current research and thank you for proposing this avenue for future exploration.

  1. I think the authors should at least outline proposed/possible reaction pathways and provide more relevant references for the sake of helping the readers.

Response: Thank you for your suggestion to outline the proposed reaction pathways and provide more relevant references. In our manuscript, we have provided a preliminary interpretation of the reaction mechanisms involved in plasma-activated water (PAW)-assisted Atomic Layer Deposition (ALD) of Al2O3. Specifically, we discuss how the surface reactions during trimethylaluminum (TMA) dosing in PAW-ALD appear analogous to those in thermal ALD. In this process, methane (CH₄) is released as a by-product when Al(CH₃)₃ chemisorbs onto the coating-substrate surface.

Furthermore, we highlight that the oxidation process in PAW-ALD is more intricate than in thermal ALD using H2O, primarily due to the influence of Reactive Oxygen and Nitrogen Species (RONS) present in PAW, especially hydrogen peroxide (H2O2) and ozone (O3). These RONS may introduce alternative oxidation pathways for –CH₃ surface groups.

Additionally, we included more references that are relevant to our discussion of these mechanisms, to provide readers with a broader context and better understanding of the results observed in PAW-ALD of Al2O3 thin films.

  1. How did the authors calculate the bandgap from XPS? In general, UV-Vis spectroscopy (optical bandgap) is required in addition to the XPS/UPS measurements to understand the band structure. But, I have not seen the relevant or related information in the manuscript.

Response: Thank you for your inquiry regarding the calculation of the bandgap from XPS measurements in our manuscript. We understand the typical approach of using UV-Vis spectroscopy in conjunction with XPS/UPS (Ultraviolet Photoelectron Spectroscopy) measurements to understand the band structure. However, in our study, we employed a method using XPS to estimate the bandgap energy.

This method is based on the well-established principle that the difference in energy between the elastic peak (e.g., oxygen peak - EO1s) and the onset of inelastic losses (Eloss) corresponds to the energy gap (Eg). To determine the bandgap energy, we performed a linear fit on the measured loss spectra curve near the elastic peak. We then subtracted the Shirley background fitting, which is the background "zero" level. The point where the linear-fit line intersects the background "zero" level indicates the onset of inelastic losses. Thus, the bandgap energy is calculated as the difference between the elastic peak energy and the onset of inelastic losses, represented by the equation Eg = Eloss - EO1s.

We acknowledge that this method might be less conventional than using UV-Vis spectroscopy for bandgap determination. However, it provides a viable alternative, especially in cases where UV-Vis spectroscopy data may not be readily available or applicable. We have included a detailed explanation of this method in the Supporting Information of our manuscript to aid in the understanding of our approach.